# Application of Synchrotron Radiation X-ray Scattering and Spectroscopy to Soft Matter

**DOI:** 10.3390/polym12071624

**Published:** 2020-07-21

**Authors:** Atsushi Takahara, Yuji Higaki, Tomoyasu Hirai, Ryohei Ishige

**Affiliations:** 1Institute for Materials Chemistry and Engineering, Kyushu University, 744 Motooka, Nishi-ku, Fukuoka 819-0395, Japan; y-higaki@oita-u.ac.jp (Y.H.); tomoyasu.hirai@oit.ac.jp (T.H.); ishige.r.aa@m.titech.ac.jp (R.I.); 2Department of Integrated Science and Technology, Faculty of Science and Technology, Oita University, 700 Dannoharu, Oita 870-1192, Japan; 3Department of Applied Chemistry, Faculty of Engineering, and Graduate School of Engineering, Osaka Institute of Technology, 5-16-1, Asahi-ku, Osaka 535-8585, Japan; 4Department of Chemical Science and Engineering, Tokyo Institute of Technology, E4-5, 2-12-1, Ookayama, Meguro-ku, Tokyo 152-8552, Japan

**Keywords:** colloidal crystal, conductive polymer, polymer brush, water, diffusion

## Abstract

Light produced by synchrotron radiation (SR) is much brighter than that produced by conventional laboratory X-ray sources. The photon energy of SR X-ray ranges from soft and tender X-rays to hard X-rays. Moreover, X-rays become element sensitive with decreasing photon energy. By using a wide energy range and high-quality light of SR, different scattering and spectroscopic methods were applied to various soft matters. We present five of our recent studies performed using specific light properties of a synchrotron facility, which are as follows: (1) In situ USAXS study to understand the deformation behavior of colloidal crystals during uniaxial stretching; (2) structure characterization of semiconducting polymer thin films along the film thickness direction by grazing-incidence wide-angle X-ray scattering using tender X-rays; (3) X-ray absorption fine structure (XAFS) analysis of the formation mechanism of poly(3-hexylthiophene) (P3HT); (4) soft X-ray absorption and emission spectroscopic analysis of water structure in polyelectrolyte brushes; and (5) X-ray photon correlation spectroscopic analysis of the diffusion behavior of polystyrene-grafted nanoparticles dispersed in a polystyrene matrix.

## 1. Introduction

X-ray scattering is a non-destructive analytical technique in which a sample is irradiated with X-rays to analyze the sample structure based on the intensity and position (angle) of the X-rays scattered by the sample. In X-ray scattering, information about small structures is obtained by analyzing the X-rays scattered at large scattering angles (wide-angle X-ray scattering: WAXS), whereas information about large structures is obtained by analyzing X-rays scattered at small scattering angles (small-angle X-ray scattering: SAXS) [1,2,3]. Therefore, by analyzing the diffraction and scattering of X-rays over a broad range of scattering angles (from small to large angles), information about polymer hierarchical structures with different length scales can be obtained. In situ structural evaluation under various mechanical stimuli is an effective method to evaluate in detail the relationship between mechanical properties and a hierarchical structure [4,5]. This is because in a polymer material, the relaxation phenomenon is remarkable, and structural changes occur after the application of a mechanical stimulus. Since the emitted synchrotron radiation (SR) light has an extremely high X-ray luminance, and an extremely short exposure time is required for data acquisition, it enables the monitoring of the time dependence of the structure under deformation and the other external perturbations [6]. High directionality of SR light is suitable for thin film analyses based on grazing incidence configuration [7]. With the high brightness of SR light and the advancement in X-ray optics, the X-ray diffraction and scattering technique using microbeam has been developed and is used by many researchers as a powerful tool for local structure analysis such as the detailed structural analysis of the internal structure of spherulites [8], and the analysis of spatial distribution of microcrystalline orientation at the carbon fiber/polymer interface [9].

The X-ray energy is chosen depending on the target experiments. So far, X-ray energy regions have been roughly divided into three regions: soft X-ray (less than 2 keV), tender X-ray (between 2 to 6 keV), and hard X-ray (higher than 6 keV). Figure 1 summarizes the X-ray scattering and spectroscopy techniques with X-rays of different energies. Until now, hard X-rays have been mainly used for scattering and diffraction. The recent emergence of high-intensity radiation sources has led to new experimental approaches. X-ray scattering and spectroscopic studies using ‘tender X-rays’, have become popular with the emergence of next-generation synchrotron radiation facilities. This is because the absorption edges and characteristic X-rays of many elements can be observed in this energy region. In the case of soft X-rays, sensitive absorption or emission phenomena, such as the electronic structure, are observed. Soft X-rays have a shorter wavelength than ultraviolet light or vacuum ultraviolet light and a longer wavelength than hard X-rays, with an energy in the range of 100–2000 eV. Unlike hard X-rays, which have a high transmittance in materials, soft X-rays have a low transmittance in materials and are easily attenuated in air. The origin is absorption by light elements present in this energy region. For example, the K absorption edge is around 540 eV for oxygen molecules and around 410 eV for nitrogen molecules. Such absorption is due to the transition of core electrons, and the low transmittance of soft X-rays reflects the strength of the interaction of X-rays with matter [10]. With regard to the penetrating power, soft X-rays below 1 keV hardly pass through a 1 cm length air, while X-rays of 10 keV transmit through more than 1 m. With a decrease in energy from tender to soft X-rays, the optical system becomes extremely difficult to design.

In this paper, we summarize our recent SR experiments with various energy regions, which are as follows: (1) in situ ultra-small-angle X-ray scattering (USAXS) study to understand the structural changes in a colloidal crystal during uniaxial stretching, (2) characterization of semiconducting polymer thin films along the film thickness direction by grazing-incidence wide-angle X-ray scattering (GIWAXS) analysis with tender X-rays, (3) X-ray absorption fine structure (XAFS) analysis of the formation mechanism of poly(3-hexylthiophene) (P3HT), (4) soft X-ray absorption and emission spectroscopy of water structure in polyelectrolyte brushes, and (5) X-ray photon correlation spectroscopy (XPCS) to analyze the diffusion behavior of polystyrene (PS)-grafted nanoparticles dispersed in a PS matrix.

## 2. Synchrotron Radiation X-ray Scattering

### 2.1. In Situ USAXS Study to Understand Structural Change in a Colloidal Crystal Elastomer During Uniaxial Stretching

Colloidal crystals of nanoparticles (NPs) are fascinating materials because of their ability to self-assemble into highly ordered periodic structures, which can be used in bottom-up technologies to fabricate photonic crystals and ordered fine pores [11]. Most of the colloidal crystals are made from hard spheres and are fragile against external perturbations because they are usually formed as a suspension in a solvent [12,13,14]. In such colloidal crystals, the repulsive force caused by the large osmotic pressure of the ionic atmospheres around the spheres is the driving force of crystallization [15]. Structures of such colloidal crystals in suspensions have been investigated with SAXS method [16,17]. On the other hand, single-component polymer-grafted nanoparticles (SPNPs), in which flexible polymer chains are densely grafted on the surface of a hard core, can form colloidal crystals without a solvent when the glass transition temperature (*T*_g_) of the grafted layer is much lower than room temperature (r.t.). This is because the densely grafted-polymer shells have an inherent large osmotic pressure below *T*_g_ and generate strong repulsive interactions [18], inducing crystallization [11,19]. Moreover, the polymer shell shows fluidity at r.t. and the colloidal crystals of SPNPs can be tougher than conventional colloidal crystals in a suspension as well as highly processable. In this section, we demonstrate the applicability of the in situ USAXS method during the mechanical deformation process through the precise analysis of structural change during the uniaxial stretching of a physically cross-linked colloidal crystal consisting of spherical SPNPs, which is a hard spherical silica (diameter: 185 nm) densely grafted with a polymer bearing hydrogen-bonding side-groups (Figure 2). The relationship between the microscopic structural change and the macroscopic stress–strain (SS) behavior is discussed based on the SS curve simultaneously acquired with USAXS patterns. The USAXS measurements were conducted mainly at BL19B2 in SPring-8 (X-ray wavelength, camera length, exposure time, and the used detector are 0.0688 nm, 41,805 mm, 3.0 s, and PILATUS-2M respectively, and detectable *q*-range = 0.005–0.2 nm^−1^) [20].

The thermally molded SPNP film exhibited a clear structural color due to the Bragg diffraction of visible light when observed from the normal direction to the film [11], which indicates that the super-lattice (colloidal crystal) responsible for the Bragg diffraction is highly oriented in the film. Therefore, the USAXS patterns of the molded SPNP film before uniaxial stretching were obtained with an X-ray beam parallel to the *xy* plane (through plane) and *xz* plane (edge plane) (Figure 3); the Cartesian coordinates (*x*, *y*, and *z*) are defined in Figure 2. In both USAXS patterns exhibited similar symmetric six diffraction peaks of which *d*-spacing is same as those expected to diffraction of close-packed structures of the SPNP (diffractions from the closest packed layer of a randomly hexagonal close-packed crystal or [111] layers of twined face-centered cubic (fcc) crystals), although the geometries of the pattern were slightly different. In the through plane, the six diffraction peaks have a six-fold symmetry; while, in the edge plane, they can be separated into four-fold symmetric diffraction peaks and two diffraction peaks on the meridian. The geometry of these diffraction peaks indicated that the close-packed lattice is oriented in the film with the close-packed layer (the [111] plane of the fcc lattice) parallel to the film surface (Figure 3c).

The molded SPNP film was then uniaxially stretched along the *x*-axis and USAXS patterns and SS curves were measured simultaneously during the stretching process (the strain rate was 10% min^−1^). The USAXS patterns of the through and edge planes at strains (*ε*) of 0, 15, 23, 30, 33, 43, and 98% are presented in Figure 3a,b, respectively (strain values are mentioned in the inset). In the close-packed layers, two lattice vectors, **a** and **b**, can be defined and the **b** vector is perpendicular to the stretching direction (Figure 3c). Before the strain reached 20% (*ε* < 20%), the diffraction pattern kept mirror symmetry about the *yz* plane, while the symmetry clearly broke after the strain exceeded 30% (*ε* > 30%). The lattice deformation was related to a macroscopic SS behavior (Figure 4a), as evident from the SS curve, which increased almost linearly before a strain of 20%, showed a decrease in slope from 20% to 40%, and gradually decreased after the strain reached 40%. In other words, the lattice deformation keeping the mirror-symmetry in the early stage (*ε* < 20%) potentially has an elastic character (the distorted lattice memorized the original close-packed structure), while the asymmetric deformation is a plastic behavior that causes energy dissipation.

The structural change was explained in detail by the model presented in Figure 3c. In the early stage (*ε* < 20%), the hexagonal shape in the close-packed layer elongated along the *x*-axis and the mirror symmetry about the *x**z* plane is maintained. In the late stage (*ε* > 30%), the SPNPs mutually slid along the *y*-axis and the mirror symmetries is broken. In these processes, the distance between the neighboring SPNPs in the *x* direction continuously increase, whereas the distance in the *y* direction does not change. This behavior is consistent with the inherent character of the densely grafted polymer layer (soft shell); that is, the strong repulsive interactions owing to the osmotic pressure occur between the densely grafted layers, which prevent the interdigitation of the soft shells on the SPNPs. As shown in Figure 3c, the SPNPs rearrange into another close-packed structure at a strain of 40%, in which the [110] plane of the fcc lattice appears parallel to the through plane. It is interesting that a close-packed lattice with different orientations from the initial state occurs without any rotation of the lattice but only with the rearrangement of the SPNPs induced by uniaxial stretching.

The lattice deformation mechanism discussed above was consistent with the change in the structural color of the SPNP film during uniaxial stretching [11]. As shown in Figure 4b, the *d*-spacing of the [111] diffraction plane of the distorted fcc lattice, *d*[111], sharply decreased before the strain reached 40% (*ε* ~ 40%). Similarly, the wavelength of the selective reflection of visible light (structural color) decreased monotonically with an increase in strain to 40% and became almost constant value after that. The decrease in the intensity of the selective reflection without a change in the peak wavelength above a strain of 30% is consistent with the change in the intensity of the [111] diffraction plane (Figure 4b). Thus, we concluded that the selective reflection was caused by the Bragg diffraction from the close-packed layer.

As mentioned in this section, the USAXS method utilizing synchrotron X-rays and having an intrinsic high intensity and collinearity with the availability of hard X-rays with high penetrability is useful for the precise structural analysis of higher-order structure formed by the hybrid materials of inorganic particles and soft polymers. This enables the in situ analysis of colloidal crystals during mechanical deformation, which can be applied to other systems with periodic structures of a few hundred nanometers and can provide essential information to understand not only equilibrium thermodynamics [21], but also non-equilibrium dynamics [22].

### 2.2. Surface Depth Profile Characterization of n-Type Semiconducting Polymer Thin Films by Grazing-Incidence Wide-Angle X-ray Scattering with Tender X-rays

Organic semiconducting materials have significant potential as flexible, low-cost, and easily processable alternatives to silicon-based semiconducting materials [23]. Polymers with perylenediimide (PDI) pendant side chains, which show crystalline states in thin films, have been paid great attention as n-type semiconducting materials [24,25,26]. The properties of their thin films are closely related to the molecular aggregation state at the surface and interface because the occupancy of surface and interface in the films increases with decreasing film thickness. To understand the importance of functional and physical properties of polymeric thin films, the precise characterization of the molecular aggregation state, which is formed in thin films normal to the film thickness, is necessary. Polymers with PDI side chains (PAc12PDI) form various kinds of crystalline structures with different orientation axes in thin films and show good performance as write-once read-many (WORM) memory and dynamic random access memory (DRAM) depending on the crystalline state [25]. However, the microcrystalline structure and its orientation behavior normal to the film thickness remain unclear.

To analyze the microcrystalline structure in PAc12DPI, we performed grazing-incidence wide-angle X-ray scattering (GIWAXS) measurements. GIWAXS is an analytical technique in which thin films are irradiated with X-ray beams with the incident angle below the critical angle, which enables the characterization of the molecular aggregation state in thin films normal to the film thickness [27,28]. The penetration depth (*Λ*) of X-rays during GIWAXS measurements depends on the X-ray energy. Figure 5 shows the relationship between *Λ* and incidence angle (*a*_i_) when GIWAXS measurements were performed using hard (0.1 nm, 12.4 keV) and tender X-rays (0.5 nm, 2.48 keV). When GIWAXS was performed using hard X-rays, the *Λ* suddenly increased when the *a*_i_ reached the critical angle, while in GIWAXS performed using tender X-rays, the *Λ* increased moderately. Hence, GIWAXS with tender X-rays enables the precise characterization of the molecular aggregation state in thin films normal to the film thickness compared with GIWAXS performed with hard X-rays. GIWAXS measurements performed with tender X-rays were carried out at the BL06 Kyushu University beamline at the SAGA Light Source (PILUTUS3 × 300K (838.8 mm × 106.5 mm with a pixel size of 172 mm × 172 mm), which can be used in high vacuum, was used in BL06. The detector can detect low energy ranging from 2 to 36 keV. BL06 can obtain X-ray energy ranging from 2.1 to 1 7 keV.). The critical angle of PAc12PDI, which is determined by theoretical calculation, was 0.4°. To understand the molecular aggregation state at the *Λ* around critical angle, *a*_i_ was set to 0.2° to 0.6°. Figure 6 shows the chemical structure of PAc12PDI and the GIWAXS patterns of PAc12PDI obtained using tender X-rays with *α*_i_ of 0.20°, 0.40°, 0.50°, and 0.60°. The penetration depths at the aforementioned *α*_i_ values were approximately 4, 8, 25, and 120 nm. The PAc12PDI thin films had two kinds of microcrystalline structures with different orientations, as shown in Figure 7.

Although the diffraction spots were broadened by the effect of limited grain sizes and statistical disorder of the lattice [30], the lattice parameters of type 1 crystal were determined as *a* =2.38 nm, *b* = 0.74 nm, *c* = 5.98 nm, and *β* = 108.13°, while those of type 2 were determined as *a* =2.38 nm, *b* = 0.74 nm, *c* = 6.00 nm, and *β* = 71.23°. Panels I–VI in Figure 7 show monoclinic lattices of type 1 and 2 with different orientations: panels I, III, and V show type 1 crystal lattices, while panels II, IV, and VI show type 2 crystal lattices. Specific diffraction spots corresponding to the [001], [100], [002], [003], and [101] planes, which can only be assigned to I, were observed at *α*_i_ of 0.20° and 0.40° (Figure 5, Figure 7). This indicates that (I) was formed from the outermost surface down to an approximately 8 nm region. The diffraction pattern of (III) was observed at 11 nm depth normal to the film thickness. The type 2 monoclinic lattices of (II) started to appear at a depth of 25 nm. Finally, all the monoclinic lattices I to VI could be seen at 50 nm depth normal to the film thickness. The population of each microcrystalline structure thus obtained is summarized in Table 1 [29].

The new GIWAXS technique using tender X-rays can be applied for the depth-resolved characterization of crystalline polymer thin films. This technique can be applied to various kinds of organic semiconducting materials, which can lead to the development of high-performance organic electronic devices.

## 3. X-ray Spectroscopy

### 3.1. Clarification of Formation Mechanism for Poly(3-hexylthiophene) (P3HT) by X-ray Absorption Fine Structure (XAFS) Measurements

P3HT is widely used in p-type organic semiconducting materials such as organic solar cells and transistors [23]. Various synthesis methods have been reported for P3HT, and P3HT with well-controlled regioregularity has been obtained by the chain-growth polycondensation method [31,32]. However, chain-growth polycondensation requires tedious polymerization steps, which limit its industrial application. On the other hand, FeCl_3_ oxidative coupling polymerization of 3HT is the simplest and most cost-effective polymerization method.

It is widely accepted that FeCl_3_ particles act as an oxidant during the oxidative coupling polymerization of 3HT [33]. Since FeCl_3_ is deactivated by humidity and has magnetic properties, determining the reaction mechanism of oxidative coupling polymerization by spectroscopic methods, including nuclear magnetic resonance (NMR) spectroscopy, is difficult. Hence, the reaction mechanism remains unclear. To overcome this problem, we prepared a homemade reaction cell that allows the reaction to proceed under inert gas conditions and monitored the reaction by X-ray absorption fine structure (XAFS) measurements performed at the BL06 Kyushu University beamline at the SAGA Light Source (Figure 8a). The Fe K-edge X-ray absorption near-edge structure (XANES) spectra of the polymerization system in CHCl_3_ obtained at different times are shown in Figure 8b. At the beginning of the reaction, the XANES spectrum shows a sharp peak at 7122 eV, which indicates that a large fraction of Fe(III) reduced to Fe(II) because of the formation of a radical cation on 3HT. As the reaction progressed, the magnitude of the peak absorption, the so-called white line, decreased, and the edge position shifted toward higher energies. This shows that the generated Fe(II) reoxidized to Fe(III). These results indicate that FeCl_3_ behaves as a catalyst rather than an oxidant [34].

To clarify the effect of solvent on the polymerization, we replaced CHCl_3_ with hexane and carried out the reaction under similar conditions. Similar to the reaction in chloroform, the peak at 7122 eV corresponding to Fe(II) was observed immediately after the addition of 3HT to the reaction mixture. The peak corresponding to Fe(II) was more pronounced than that observed for CHCl_3_ and did not decrease with time. This indicates that the oxidation of Fe(II) to Fe(III) does not occur in hexane. Hence, we concluded that the oxidation is a solvent-dependent process. Figure 8c shows the most plausible reaction mechanism for the oxidative coupling polymerization reaction of 3HT as observed from the XAFS spectra. Hence, XAFS measurements allow the analysis of complicated reaction mechanisms of various chemical reactions [34,35].

### 3.2. Analysis of Hydrogen-Bonded Network of Water Confined in Polyelectrolyte Brushes by Soft X-ray Absorption and Emission Spectroscopies

The molecular-level understanding of the hydrogen-bonded network structure of water in crowding fields is a ubiquitous and crucial issue in both chemistry and biology because biological processes occur in confined cells and the fate of the process depends on the molecular interactions involving water [36]. The network structure of water in artificial confinement fields including reverse micelles [37], carbon nanotubes [38], and metal–organic frameworks [39] was studied by infrared and Raman spectroscopies, while the water structure in the vicinity of surfaces was determined by means of surface force apparatus, frequency modulation scanning probe microscopy [40], and sum frequency generation spectroscopy [41]. In most cases, water produces a strongly hydrogen-bonded network in confinement fields; however, the configuration of water structure in the crowded environment of a living cell is still being debated and has not been elucidated yet. Since several charged biomolecules—in particular, proteins, nucleic acids, and sugars—are present in the confinement fields, the configuration of water structure confined in the fields composed of polyelectrolytes is worth understanding.

We investigated the unique water structure confined in polyelectrolyte brushes by synchrotron-based soft X-ray absorption and emission spectroscopies (XAS and XES, respectively) [42]. XAS and XES yield element-specific unoccupied and occupied electronic structures to show the local symmetry of hydrogen bonding. The detailed analyses of distinct electronic structures based on the XAS and XES spectra provide constructive information about the hydrogen-bonded network of water [43,44,45,46,47]. Polyelectrolyte brushes are thin films composed of surface-tethered polyelectrolyte chains. By using the surface-initiated atom transfer radical polymerization (SI-ATRP) technique to prepare polymer brushes, the lateral chain density can be controlled to give concentrated polymer brushes [48,49]. The polyelectrolyte brush chains spontaneously incorporate water upon exposure to humid vapor due to charge–dipole interactions. Thus, the XAS and XES spectra of the confined water molecules in a confined field of crowded polyelectrolyte chains can be acquired.

A positively charged poly(2-(methacryloyloxy)ethyl trimethylammonium chloride) (PMTAC) brush was prepared by SI-ATRP on a 150 nm-thick SiC membrane with an Au layer on the top. XAS and XES spectroscopies were performed at the BL07LSU HORNET station in SPring-8 [50]. The polymer brush was exposed to water vapor to incorporate water into the bulk of the brush. The O_1s_ XES spectrum of water confined in the PMTAC brush was compared with those of the dry PMTAC brush in vacuum, bulk water, and ice. Two peaks assigned to 1b_1_’ and 1b_1_’’ in the O_1s_ XES spectrum provide insights into the hydrogen-bonding network in liquid water because the fully hydrogen-bonded ice gives rise to the 1b_1_’ peak, while the less hydrogen-bonded water gives rise to the 1b_1_’’ peak. The water confined in the PMTAC brush almost does not show the 1b_1_’’ peak in the XES spectrum, which implies that the confined water is highly hydrogen-bonded similar to ice even at room temperature (Figure 9a). In addition, the slight enhancement of the 3a_1_ peak indicates the distortion of hydrogen bonds. In the XAS spectrum of water confined in the PMTAC brush, the pre-edge absorption peak at 535 eV disappeared and the post-edge absorption peak shifted toward lower energies, which indicates slightly distorted but somewhat ordered hydrogen bonds (Figure 9b). Thus, the XES and XAS results are consistent and reveal the tetrahedrally coordinated strong hydrogen-bonding configuration with a slight distortion of water confined in the PMTAC brush. Besides, the results are consistent with the water structure outside the water droplet on the PMTAC brush determined by high-spatial resolution infrared spectroscopy [51]. The structured water would prevent the complete wetting of the extremely hydrophilic polyelectrolyte brush.

### 3.3. Analysis of Diffusion Behavior of Polystyrene-Grafted Nanoparticles in a Polystyrene Matrix by X-ray Photon Correlation Spectroscopy

The study of the dynamical behavior of colloids or NPs is considerably important for nanomedicine such as drug delivery systems. Although the dynamical behavior of NPs in polymer matrices is different from that of the normal Brownian motion, the details of such dynamical behavior are still unknown. Moreover, the dynamical features of polymer-grafted NPs are rarely studied. X-ray photon correlation spectroscopy (XPCS) uses partially coherent X-rays to provide experimental access to a variety of microscopic dynamic phenomena [52]. Figure 10 shows the schematic representation of the principle of XPCS. If a random arrangement of scatterers is illuminated by coherent radiation, the scattered intensity exhibits a so-called speckle pattern that reflects the instantaneous configuration of the scatterers. The movement of scatterers changes the speckle pattern, which provides information about the dynamics of the system. The dynamics of PS-grafted NPs dispersed in a PS matrix were studied by XPCS.

Silica NPs (110 nm in diameter) grafted with PS brushes (M_n_ = 2.30 × 10^4^, M_w_/M_n_ = 1.41, and 0.28 chains nm^-2^) were dispersed in a PS matrix. To investigate the effect of the chain length of the polymer matrix, two different samples were prepared: one in which the molecular weight of the matrix (M_n_ = 1.05 × 104, M_w_/M_n_ = 1.09, and *T*_g_ = 367 K) was smaller than that of the brushes (sample-A), and another in which the molecular weight of the matrix (M_n_ = 3.00 × 10^4^, M_w_/M_n_ = 1.06, and *T*_g_ = 374 K) was larger than that of the brushes (sample-B).

An XPCS instrument equipped with a fast pixel array detector PILATUS with a grid mask resolution enhancer has been installed at the BL19LXU beamline of SPring-8 [53]. For XPCS measurements, samples are coherently illuminated, and the fluctuation in the scattered speckle intensity is observed. The time-autocorrelation functions, g_2_(q,t), are then calculated from the fluctuation, where q and t are the wave vector and time, respectively.

Figure 11 shows the representative g_2_(q,t) for sample-A at q = 2.15 × 10^−2^ nm^−1^ in the investigated temperature (*T*) range of 433–503 K. The solid lines are the best-fit curves with the stretched (or compressed) exponential equation, g_2_(q,t)= A exp[−2(Γt)^β^]+ 1, where A, Γ, and β (≠1) are the contrast, relaxation rate, and stretched (or compressed) exponent, respectively. Here, β ≠ 1 implies that the NP behavior is different from normal Brownian motion.

To understand the NP behavior on a microscopic scale, the experimental data were analyzed with a kind of continuous time random walk (CTRW) model [54]. In the CTRW model, the particle motion is expressed by discrete steps. The mean displacement of a particle after N steps is expressed as Nαδ, and the mean elapsed time as Nτ_0_. Here, δ is the average length of a single jump and τ_0_ is the mean time between the jumps. Brownian motion is described by α=0.5, where α > 0.5 and α < 0.5 correspond to hyperdiffusive and subdiffusive behaviors, respectively. In the fitting analysis of the measured g2(q,t) with the CTRW model at each temperature, although α = 0.8 was obtained at T = 433 K, α decreased with increasing temperature to less than 0.5 at T ≈ 457 K, which is 1.25Tg, and finally dropped to α = 0.3 at *T* = 503 K. Thus, the NP behavior of sample-A changed from hyperdiffusive motion to subdiffusive motion at 1.25Tg [55].

On the other hand, in the case of sample-B, although α > 0.5 was obtained at *T* < 1.25*T*_g_, α = 0.5, corresponding to the normal Brownian motion, was obtained at *T* > 1.25*T*_g_. The different dynamical behaviors of the two samples at *T* > 1.25*T*_g_ arise from the difference in the chain length of the matrix. This may be explained as follows: in the case of sample-A, since the chains of the matrix are shorter than that of the brushes, the brushes became wet; hence, the interaction between the matrix chains and brushes caused the subdiffusive motion of the NPs. In contrast, in the case of sample-B, the longer matrix chains do not wet the brushes; therefore, the considerably lesser interaction between the matrix and brushes allowed the NPs to move by normal Brownian motion. This XPCS technique has been applied to polymer thin films by grazing incidence mode [46].

## 4. Conclusions

We described our recent research on the structural analysis of soft materials by synchrotron radiation X-ray scattering and spectroscopy. Synchrotron radiation X-rays are used not only for diffraction and scattering, but also for spectroscopic measurements that reflect the electronic state. However, their application to soft materials is lagging behind that of metals and inorganic materials. In future, when a third-generation radiation source is built, it is expected that high-quality light will be provided from the tender to the soft X-ray range. Then, the structure and properties of thin soft matter materials under external stimuli can be characterized in detail by scattering measurements when the contrast between the elements are tunable. Also, if coherent tender-to-soft X-ray beams are obtained, chemical XPCS might be realized and reveal various element specific interactions of functional soft materials.

## Figures and Tables

**Figure 1 polymers-12-01624-f001:**
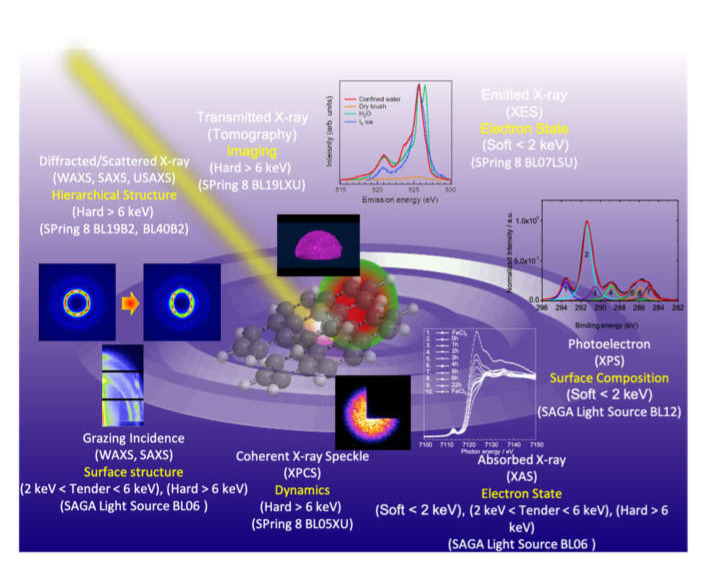
Schematic representation of X-ray scattering and spectroscopy techniques with X-rays of different energies.

**Figure 2 polymers-12-01624-f002:**
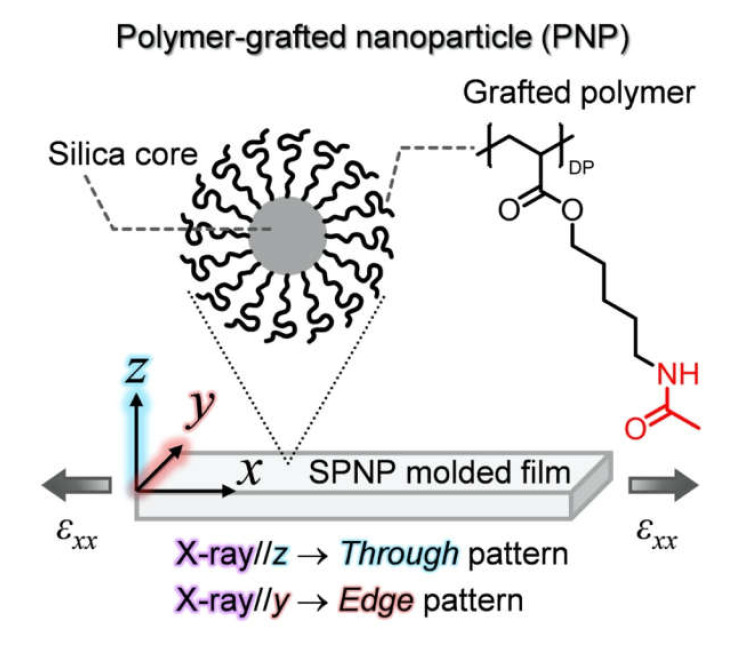
Illustration of the single-component polymer-grafted nanoparticle (SPNP) molded thin film and the chemical structure of the grafted polymer. The *x*, *y*, and *z* axes in the Cartesian coordinates are defined as the stretching direction, width direction, and thickness direction of the film, respectively. The stretching direction is indicated by the thick arrows (elongation stain *ε**_x_**_x_*)). The X-ray beam is irradiated perpendicular to the through (*xy*) and edge (*xz*) planes, respectively. Reproduced with permission [19]. Copyright 2016, IUCr.

**Figure 3 polymers-12-01624-f003:**
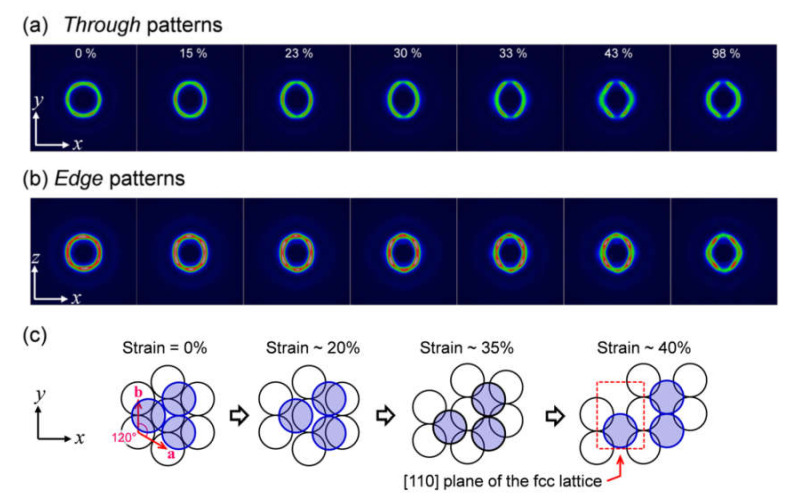
USAXS patterns of the through plane (**a**) and edge plane (**b**). The model of the deformation mechanism of the SPNP lattice is presented in panel (**c**), in which the close-packed layers lie on the through plane at the initial state. The lattice vectors of the two-dimensional hexagonal lattice in the close-packed plane are represented by **a** and **b** in panel (**c**). Reproduced with permission [19]. Copyright 2016, IUCr.

**Figure 4 polymers-12-01624-f004:**
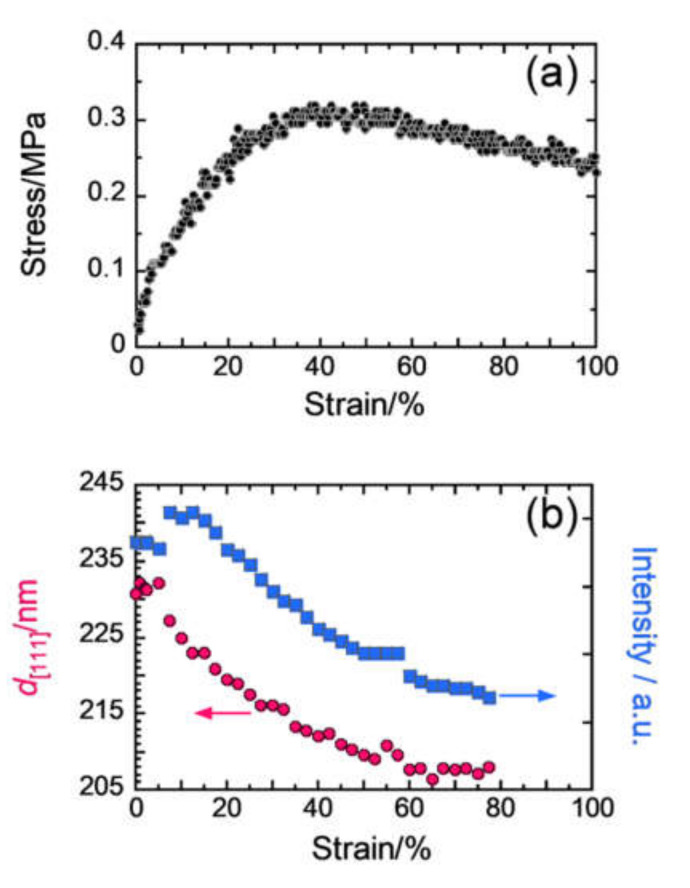
Stress–strain curve obtained during in situ USAXS measurement (**a**). (**b**) Strain dependence of the *d*-spacing of the [111] plane of the deformed fcc lattice, *d*[111], and the intensity of *d*[111] diffraction. Reproduced with permission [19]. Copyright 2016, IUCr.

**Figure 5 polymers-12-01624-f005:**
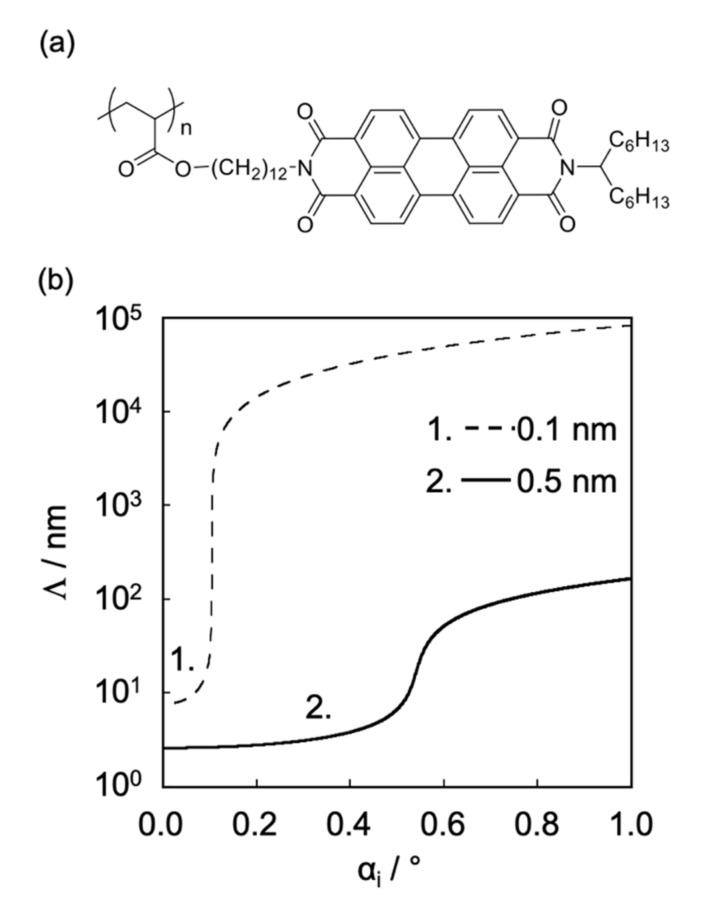
(**a**) Chemical structure of PAc12PDI. (**b**) X-ray penetration depth profiles of PAc12PDI thin films using hard (0.1 nm, 12.4 keV) and tender X-rays (0.5 nm, 2.48 keV). Reproduced with permission [29]. 2018, American Chemical Society.

**Figure 6 polymers-12-01624-f006:**
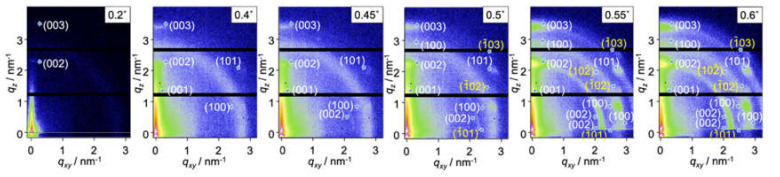
GIWAXS patterns of a PAc12DPI thin film on a Si wafer acquired using tender X-ray beam with different incident angles from 0.2 to 0.6°. Reproduced with permission [29]. 2018, American Chemical Society.

**Figure 7 polymers-12-01624-f007:**
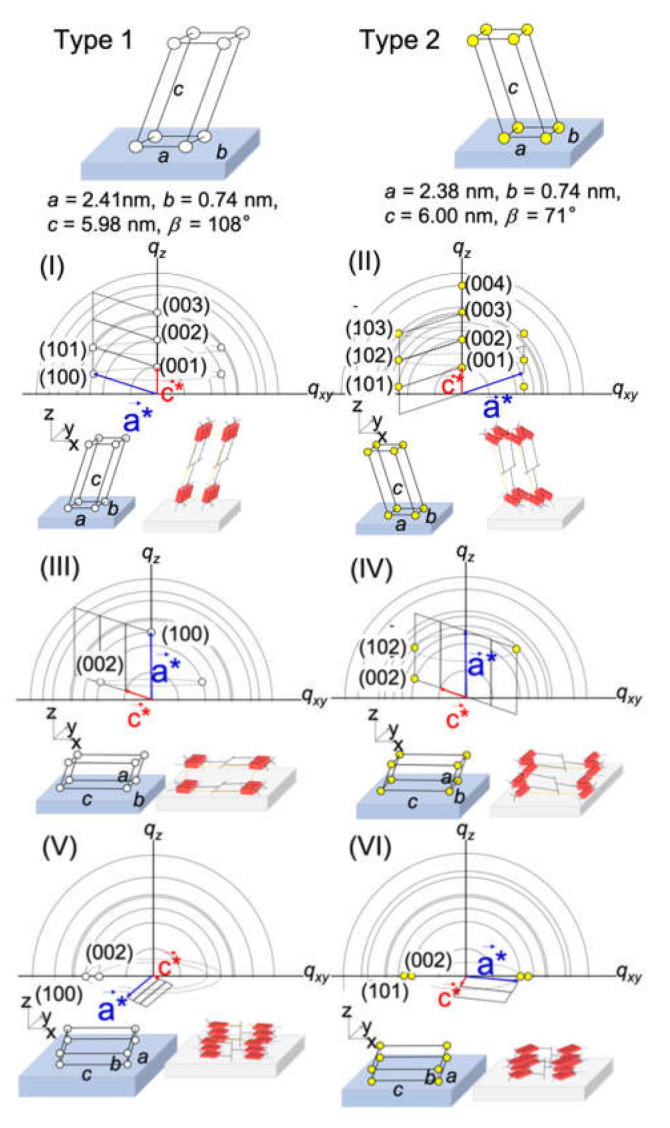
Two kinds of crystalline lattices with different orientations. (**I**), (**III**), and (**V**) are Type 1, while (**II**), (**IV**), and (**VI**) are Type 2. Reproduced with permission [29]. 2018, American Chemical Society.

**Figure 8 polymers-12-01624-f008:**
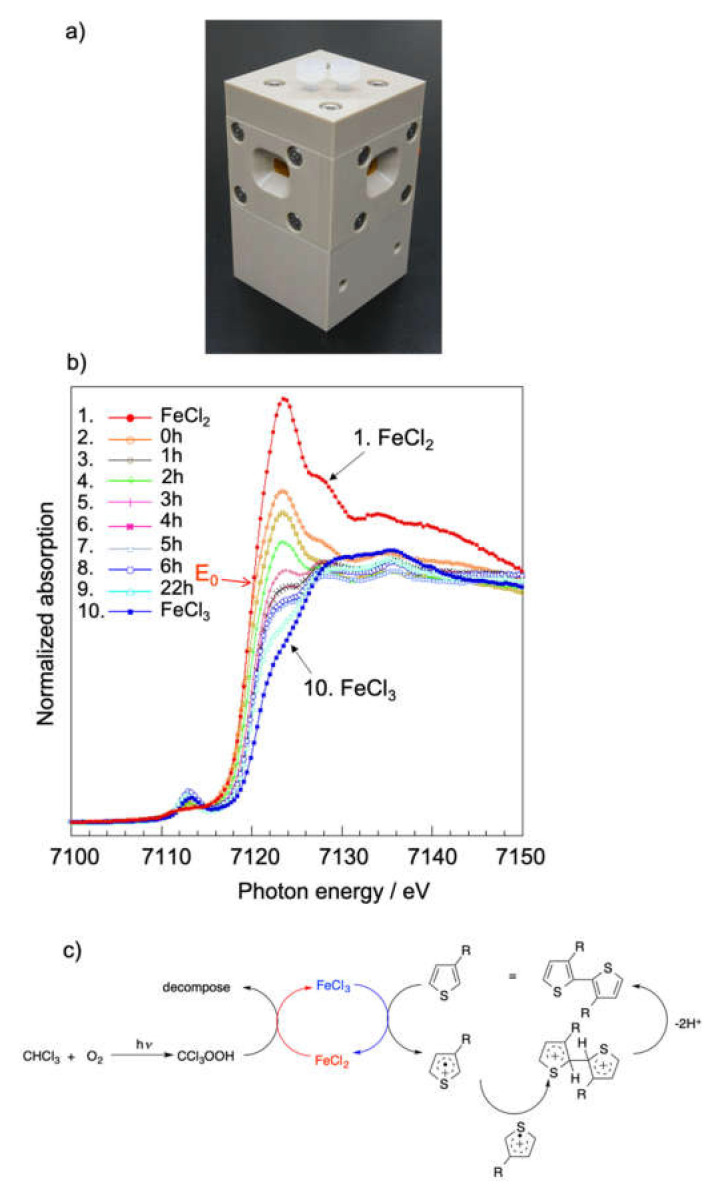
(**a**) Homemade cell for XAFS measurements. (**b**) XANES spectra of the polymerization of 3HT with FeCl_3_ in CHCl_3_. (**c**) Reaction mechanism of oxidative coupling polymerization of 3HT. Reproduced with permission [34]. 2015, John Wiley-VCH.

**Figure 9 polymers-12-01624-f009:**
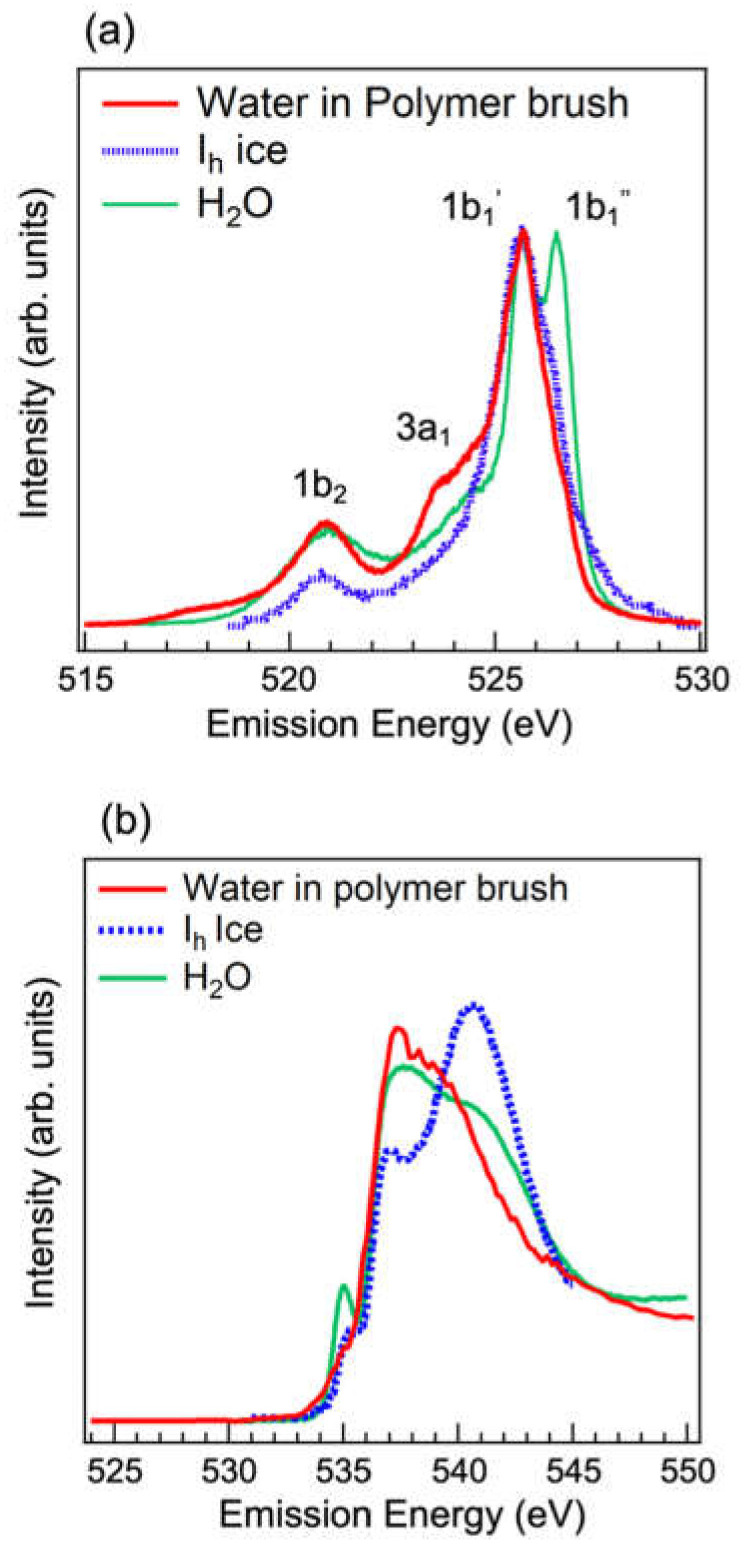
(**a**) O 1s soft X-ray emission spectra of water confined in the polyelectrolyte brush, the dry brush measured in vacuum, liquid H_2_O, and ice *I*_h_. The excitation energy is 550.3 eV, which is much higher than the ionization threshold. (**b**) Soft X-ray absorption spectra of water confined in the polyelectrolyte brush, dry brush measured in vacuum, liquid H_2_O at room temperature, and *I*_h_ ice. Reproduced with permission [42]. 2017, American Chemical Society.

**Figure 10 polymers-12-01624-f010:**
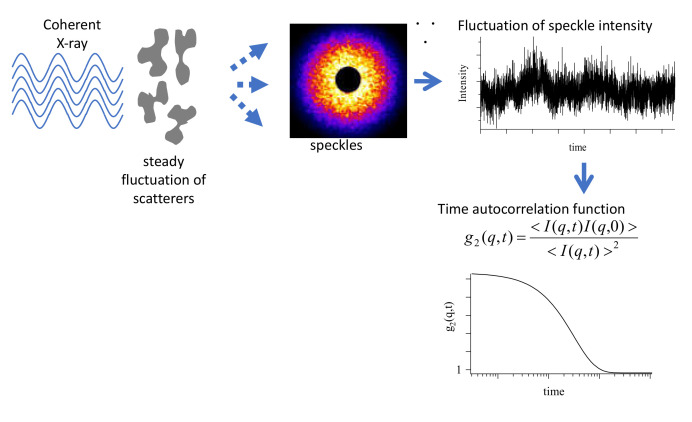
Schematic representation of principle of XPCS measurement.

**Figure 11 polymers-12-01624-f011:**
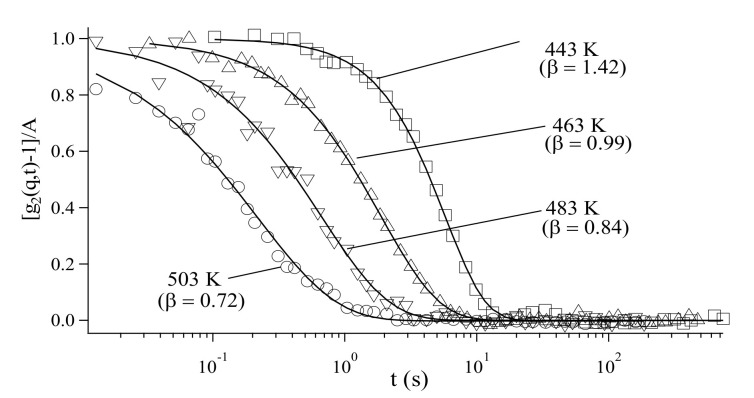
Representative results of normalized time-autocorrelation functions at *q* = 2.15 × 10^−2^ nm^−1^ of the scattered speckle intensity from PS-grafted NPs in a PS matrix of sample-A at different temperatures. The solid lines are the fitted curves by stretched exponential function. Reproduced from [55], with permission from the American Physical Society.

**Table 1 polymers-12-01624-t001:** Population of each microcrystalline structure at a given *α*_i_.

Incidence Angle °/Penetration Depth nm	(I)/%	(II)/%	(III)/%	(IV)/%	(V)/%	(VI)/%
**0.2°/4 nm**	100	0.0	0.0	0.0	0.0	0.0
**0.4°/8 nm**	100	0.0	0.0	0.0	0.0	0.0
**0.45°/11 nm**	93.6	0.0	6.4	0.0	0.0	0.0
**0.5°/25 nm**	84.3	1.9	13.8	0.0	0.0	0.0
**0.55°/50 nm**	56.1	2.9	3.5	0.7	26.4	10.5
**0.6°/120 nm**	17.8	1.0	2.4	0.8	49.6	28.4

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
