# Peer review of "Application of Synchrotron Radiation X-ray Scattering and Spectroscopy to Soft Matter"

_polymers, 2020, doi:10.3390/polym12071624_

Round 1

Reviewer 1 Report

General comments

The authors have produced a nice overview of the techniques that are available for soft condensed matter at synchrotron beamlines.  Always in such reviews there is a fine balance between details and maintaining an accessible coherent story.  I do not believe as is, this balance is achieved in this manuscript, since the introductory section does leave the reader with sufficient information to appreciate the article as a whole.  As a frequent synchrotron user I was impressed by the case studies that have been selected which add to an interesting narrative, however the level of information in the introduction is not sufficient.  This situation could be easily resolved with simple statements about the capabilities of beamlines to measure the elastically scattered intensity of X-rays as a function of wavelength (WAXS and SAXS), as well as energy dependent intensities on a 2D detector.  There are some important combination of the two approaches using energy dispersive techniques.

While the English is a little awkward, overall its meaning is clear but I have the following suggestions to add to the clarity of the document:

Specific comments

Line 34 analyzing should be analyze

Figure 1 would benefit from labelling with the energy ranges for each technique.

Line 90 The phrase “suspension of a fluidic solvent” is confusing for me.  Wouldn’t a “suspension in a solvent” be simpler, or is there a subtlety that I am missing?

Section 2.1

Figure 2 would benefit from some labelling of the measurement geometries for scattering patterns shown in Figure 3.

Line 115 “normal direction of the film” should be “normal direction to the film”

Line 116 “related to the Bragg diffraction” should be “responsible for the Bragg diffraction”

Line 122 “(diffractions” should be ”(diffraction”

Line 123 “the six diffractions” should be “the six diffraction peaks”

Line 124 “four-fold symmetric diffractions” should be “four-fold symmetric diffraction peaks” and “two diffractions on” should become “two diffraction peaks on”

It would have been nice to have some brief description of the beamline setup and the range of scattering vectors that were covered.

Section 2.2

Line 203 I believe a short explanation of the critical angle would enhance the discussion.  The use of tender X-rays in resonant X-ray reflectivity experiments would have been entirely appropriate in this case.

Line 211 “along film thickness direction”.  From my understanding of the text the depth of penetration is modulated by the incident angle, then this should become “normal to the film thickness”

There could be a mention of the use of the broadening of Bragg peaks to examine domain sizes, e.g. Smilgies, D. M. (2009). "Scherrer grain-size analysis adapted to grazing-incidence scattering with area detectors." Journal of Applied Crystallography 42: 1030-1034.

Author Response

Reviewer #1

Comment 1:
The authors have produced a nice overview of the techniques that are available for soft condensed matter at synchrotron beamlines.  Always in such reviews there is a fine balance between details and maintaining an accessible coherent story.  I do not believe as is, this balance is achieved in this manuscript, since the introductory section does leave the reader with sufficient information to appreciate the article as a whole.  As a frequent synchrotron user I was impressed by the case studies that have been selected which add to an interesting narrative, however the level of information in the introduction is not sufficient.  This situation could be easily resolved with simple statements about the capabilities of beamlines to measure the elastically scattered intensity of X-rays as a function of wavelength (WAXS and SAXS), as well as energy dependent intensities on a 2D detector.  There are some important combination of the two approaches using energy dispersive techniques.

Response:

Thank you very much for your kind review. In order to better understand for readers, we modified Figure 1 and added more concise introduction with some new reference. The information for beam line specification including wavelength and detectors was added to each section. We added above sentence in the main text as shown in red color.  

Comment 2:
Line 34 analyzing should be analyze

Response:
Thank you very much for your constructive suggestion. We replaced “analyzing” for “analyze”.

Comment 3:
Figure 1 would benefit from labelling with the energy ranges for each technique.

Response:
This comment is related to comment 1. We have added energy that is utilized for each X-ray experiments in Figure1. Also we have added information of beam line for each experiments.

Comment 4:
Line 90 The phrase “suspension of a fluidic solvent” is confusing for me. Wouldn’t a “suspension in a solvent” be simpler, or is there a subtlety that I am missing?

Response:
Thank for your suggestion. We replaced “suspension of a fluidic solvent” for “suspension in a solvent”.

Comment 5:
Figure 2 would benefit from some labelling of the measurement geometries for scattering patterns shown in Figure 3.

Response:

Thank you very much for your constructive suggestion. We have imbedded new Figure 2 including labelling of the geometries for the scattering patterns.  

Comment 6:
Line 115 “normal direction of the film” should be “normal direction to the film”

Response:
Thank for your suggestion. We have replaced “normal direction of the film” for “normal direction to the film”.

Comment 7:
Line 116 “related to the Bragg diffraction” should be “responsible for the Bragg diffraction”

Response:

Thank for your suggestion. We replaced “related to the Bragg diffraction” for “responsible for the Bragg diffraction”.

Comment 7:
Line 122 “(diffractions” should be ”(diffraction)”

Response:

Thank for your suggestion. We replaced “diffractions” for “diffraction”.

Comment 8:
Line 123 “the six diffractions” should be “the six diffraction peaks”

Response:

Thank for your suggestion. We replaced “the six diffractions” for “the six diffraction peaks”.

Comment 9:
Line 124 “four-fold symmetric diffractions” should be “four-fold symmetric diffraction peaks” and “two diffractions on” should become “two diffraction peaks on”

Response:

Thank for your suggestion. We replaced “four-fold symmetric diffractions” for “our-fold symmetric diffraction peaks” and “two diffractions on” for “two diffraction peaks on”, respectively .

Comment 10:
It would have been nice to have some brief description of the beamline setup and the range of scattering vectors that were covered.

Response:

Thank you very much for your constructive suggestion. We have added following sentence in the main text as shown in red color. X-ray wavelength, camera length, exposure time, and the used detector are 0.0688 nm, 41,805 mm, 3.0 s, and PILATUS-2M respectively, and detectable q-range = 0.005–0.2 nm−1.

Comment 11:
Line 203 I believe a short explanation of the critical angle would enhance the discussion.  The use of tender X-rays in resonant X-ray reflectivity experiments would have been entirely appropriate in this case.

Response:

The critical angle of PAc12PDI was estimated on the basis of theoretical calculation. The value was 0.4°. We added following sentence in the main text as red color. The critical angle of PAc12PDI, which is determined by theoretical calculation, was 0.4°. To understand the molecular aggregation state at the L around critical angle, ai was set to 0.2° to 0.6°.

On the other hand, we did not direct evaluate critical angle using X-ray reflectivity. No action was taken for this part in this review.

Comment 12:
Line 211 “along film thickness direction”.  From my understanding of the text the depth of penetration is modulated by the incident angle, then this should become “normal to the film thickness”

Response:

Thank for your suggestion. We replaced “along film thickness direction” for “normal to the film thickness”

Comment 13:
There could be a mention of the use of the broadening of Bragg peaks to examine domain sizes, e.g. Smilgies, D. M. (2009). "Scherrer grain-size analysis adapted to grazing-incidence scattering with area detectors." Journal of Applied Crystallography 42: 1030-1034.

Response:
Thank you very much for your constructive suggestion. We have added following sentence and the references as ref. 30 in the main text as shown in red color.  Although the diffraction spots were broadened by the effect of limited grain sizes and statistical disorder of the lattice[30], the lattice parameters of type 1 crystal were determined as a =2.38 nm, b = 0.74 nm, c = 5.98 nm, and b = 108.13°, while those of type 2 were determined as a =2.38 nm, b = 0.74 nm, c = 6.00 nm, and b = 71.23°.

Reviewer#2

Comment 1:
The manuscript by Takahara et al. reports on the interest of different X-ray scattering and spectroscopy techniques in the context of soft matter study. Five different techniques are described in the review paper that are illustrated with examples of the authors (USAXS, GiSAXS, XANES, XES and XPCS). Overall, the manuscript is well written with enough figures and illustrations for each case. Some efforts have been made in the introduction and in the opening paragraphs of each example to explain each technique, but this could be improved by making a separate paragraph (one for SR X-ray scattering and one for SR X-ray absorption) before introducing the different studies. If possible, the pros and cons of each technique as well as some referenced papers of each beamline (if any) should be provided to strengthen the interest for a broad audience. I recommend publishing this paper after careful revision based on the above comment and the suggestions/remarks below:

Response:

Thank you very much for your constructive suggestion. We have added beamline specification and several references in the main text as shown in red color.  

Comment 2:
A large part of the Soft Matter community use the term WAXS instead of WAXD. What is the difference between diffraction and scattering of X-rays?

Response:

We divided diffraction and scattering in the previous one. But as reviewer mentioned, there are not big differences between WAXD and WAXS, because diffraction patterns of soft matters always include not only diffraction spots but also broadened scatterings characteristic to the disordered structures. Moreover, WAXS is currently widely utilized term in the X-ray scattering fields. We replaced “WAXD” for “WAXS”.

Comment 3:
 (Line 56) “high-energy hard X-rays”: what is the range?

Response:

Thank you very much for your suggestion. The term of high is not suitable word because we did not use real high energy range so far. We removed the word “high” from the main text.

Comment 4:
Beyond the structural change, USAXS and SAXS techniques give access to the size and the concentration of scattered nanoparticles (see for instance Maes et al., Chem. Mater., 2018, 30, 3952; Li et al., Chem. Rev., 2016, 116, 11128). Such works could be mentioned in the introduction or in an opening paragraph about scattering techniques.

Response:

Thank you very much for your suggestion. We have added references as ref. 16 and 17 in the main text as shown in red color.

Comment 5:
(Lines 96 and 152) “strong repulsive interactions”: what is the magnitude of these repulsive interactions in such systems?

Response:

The comment makes good point and the magnitude of the repulsive force is important to consider the characteristics of the condensed polymer brushes (CPB). However, the repulsive force of CPB layers strongly depends on the distance between particles and it is difficult to say a specific value. What we would like to remark in this sentence is the repulsive force exceeds attractive interaction between the CPB layers rather than a specific value of the repulsive force. In this regard, however, Fukuda and Tsujii et al investigated compressibility of CPB of PMMA by using AFM technique and reported that the repulsive force steeply increased from 0.001 to 10 mN/m when CPB was compressed 20% of the original thickness in toluene (Shinpei Yamamoto, Muhammad Ejaz, Yoshinobu Tsujii, and Takeshi Fukuda,Macromolecules 2000, 33, 5608-5612). For general readers easily access a specific value observed in the PMMA CPB, we cited the paper in this sentence.

Comment 6:
(Line 184): A scheme of perylenediimide should be provided, typically in figure 5

Response:

As reviewer mention, we have to put chemical structure of PAc12PDI to better understanding for readers. We have put chemical structure in the main text.

Comment 7:
(Line 235): Please add some references about the use of GIWAXD for other polymer thin films

Response:

Thank you very much for your suggestion. We have added references as ref. 28 and 29 in the main text as shown in red color.

Comment 8:
The positions of Fe3+ and Fe2+ should be highlighted in Fig.8b. Have any simulations been run on these data (curve fitting from references, MCR-ALS chemometry…)?

Response:

Thank you very much for your suggestion. We have highlighted Fe3+ and Fe2+ in Figure 8b.

On the other hand, we have not use any simulation to fit the spectra so far. No action was taken for this part.

Comment 9:
The conclusions could be further developed by summarizing the benefits of using SR but also the limits for soft condensed matter (what about the beam damage?), and by providing an opinion on possible future developments and perspectives of using large-scale facilities in this domain.

Response:

Thank you very much for your suggestion. Beam damage has also investigated by comparing photo count on before and after evaluated region using synchrotron beam with tender and hard X-rays. Basically, the photo count number is comparable between before and after measurements. Hence, we concluded that we can ignore the beam damage. No action was taken for this part.

On the other hand, as reviewer mentioned, conclusion should be modified. We put following sentence in the main text as shown in red color. Then, the structure and properties of thin soft matter materials under external stimuli can be characterized  in detail by scattering measurements when the contrast between the elements are tunable.  Also, if the coherent tender to soft X-ray beams are obtained chemical XPCS might be realized and reveal various element specific interaction of functional soft materials.

Reviewer 2 Report

The manuscript by Takahara et al. reports on the interest of different X-ray scattering and spectroscopy techniques in the context of soft matter study. Five different techniques are described in the review paper that are illustrated with examples of the authors (USAXS, GiSAXS, XANES, XES and XPCS). Overall, the manuscript is well written with enough figures and illustrations for each case. Some efforts have been made in the introduction and in the opening paragraphs of each example to explain each technique, but this could be improved by making a separate paragraph (one for SR X-ray scattering and one for SR X-ray absorption) before introducing the different studies. If possible, the pros and cons of each technique as well as some referenced papers of each beamline (if any) should be provided to strengthen the interest for a broad audience. I recommend publishing this paper after careful revision based on the above comment and the suggestions/remarks below:

  1. A large part of the Soft Matter community use the term WAXS instead of WAXD. What is the difference between diffraction and scattering of X-rays?
  2. (Line 56) “high-energy hard X-rays”: what is the range?
  3. Beyond the structural change, USAXS and SAXS techniques give access to the size and the concentration of scattered nanoparticles (see for instance Maes et al., Chem. Mater., 2018, 30, 3952; Li et al., Chem. Rev., 2016, 116, 11128). Such works could be mentioned in the introduction or in an opening paragraph about scattering techniques.
  4. (Lines 96 and 152) “strong repulsive interactions”: what is the magnitude of these repulsive interactions in such systems?
  5. (Line 184): A scheme of perylenediimide should be provided, typically in figure 5
  6. (Line 235): Please add some references about the use of GIWAXD for other polymer thin films
  7. The positions of Fe3+ and Fe2+ should be highlighted in Fig.8b. Have any simulations been run on these data (curve fitting from references, MCR-ALS chemometry…)?
  8. The conclusions could be further developed by summarizing the benefits of using SR but also the limits for soft condensed matter (what about the beam damage?), and by providing an opinion on possible future developments and perspectives of using large-scale facilities in this domain.

Author Response

Reviewer#2

Comment 1:
The manuscript by Takahara et al. reports on the interest of different X-ray scattering and spectroscopy techniques in the context of soft matter study. Five different techniques are described in the review paper that are illustrated with examples of the authors (USAXS, GISAXS, XANES, XES and XPCS). Overall, the manuscript is well written with enough figures and illustrations for each case. Some efforts have been made in the introduction and in the opening paragraphs of each example to explain each technique, but this could be improved by making a separate paragraph (one for SR X-ray scattering and one for SR X-ray absorption) before introducing the different studies. If possible, the pros and cons of each technique as well as some referenced papers of each beamline (if any) should be provided to strengthen the interest for a broad audience. I recommend publishing this paper after careful revision based on the above comment and the suggestions/remarks below:

Response:

Thank you very much for your constructive suggestion. We have added beamline specification and several references in the main text as shown in red color.  

Comment 2:
A large part of the Soft Matter community use the term WAXS instead of WAXD. What is the difference between diffraction and scattering of X-rays?

Response:

We divided diffraction and scattering in the previous one. But as reviewer mentioned, there are not big differences between WAXD and WAXS, because diffraction patterns of soft matters always include not only diffraction spots but also broadened scatterings characteristic to the disordered structures. Moreover, WAXS is currently widely utilized term in the X-ray scattering fields. We replaced “WAXD” for “WAXS”.

Comment 3:
 (Line 56) “high-energy hard X-rays”: what is the range?

Response:

Thank you very much for your suggestion. The term of high is not suitable word because we did not use real high energy range so far. We removed the word “high” from the main text.

Comment 4:
Beyond the structural change, USAXS and SAXS techniques give access to the size and the concentration of scattered nanoparticles (see for instance Maes et al., Chem. Mater., 2018, 30, 3952; Li et al., Chem. Rev., 2016, 116, 11128). Such works could be mentioned in the introduction or in an opening paragraph about scattering techniques.

Response:

Thank you very much for your suggestion. We have added references as ref. 16 and 17 in the main text as shown in red color.

Comment 5:
(Lines 96 and 152) “strong repulsive interactions”: what is the magnitude of these repulsive interactions in such systems?

Response:

The comment makes good point and the magnitude of the repulsive force is important to consider the characteristics of the condensed polymer brushes (CPB). However, the repulsive force of CPB layers strongly depends on the distance between particles and it is difficult to say a specific value. What we would like to remark in this sentence is the repulsive force exceeds attractive interaction between the CPB layers rather than a specific value of the repulsive force. In this regard, however, Fukuda and Tsujii et al investigated compressibility of CPB of PMMA by using AFM technique and reported that the repulsive force steeply increased from 0.001 to 10 mN/m when CPB was compressed 20% of the original thickness in toluene (Shinpei Yamamoto, Muhammad Ejaz, Yoshinobu Tsujii, and Takeshi Fukuda,Macromolecules 2000, 33, 5608-5612). For general readers easily access a specific value observed in the PMMA CPB, we cited the paper in this sentence.

Comment 6:
(Line 184): A scheme of perylenediimide should be provided, typically in figure 5

Response:

As reviewer mention, we have to put chemical structure of PAc12PDI to better understanding for readers. We have put chemical structure in the main text.

Comment 7:
(Line 235): Please add some references about the use of GIWAXD for other polymer thin films

Response:

Thank you very much for your suggestion. We have added references as ref. 28 and 29 in the main text as shown in red color.

Comment 8:
The positions of Fe3+ and Fe2+ should be highlighted in Fig.8b. Have any simulations been run on these data (curve fitting from references, MCR-ALS chemometry…)?

Response:

Thank you very much for your suggestion. We have highlighted Fe3+ and Fe2+ in Figure 8b.

On the other hand, we have not use any simulation to fit the spectra so far. No action was taken for this part.

Comment 9:
The conclusions could be further developed by summarizing the benefits of using SR but also the limits for soft condensed matter (what about the beam damage?), and by providing an opinion on possible future developments and perspectives of using large-scale facilities in this domain.

Response:

Thank you very much for your suggestion. Beam damage has also investigated by comparing photo count on before and after evaluated region using synchrotron beam with tender and hard X-rays. Basically, the photo count number is comparable between before and after measurements. Hence, we concluded that we can ignore the beam damage. No action was taken for this part.

On the other hand, as reviewer mentioned, conclusion was modified. We put following sentence in the main text as shown in red color. Then, the structure and properties of thin soft matter materials under external stimuli can be characterized  in detail by scattering measurements when the contrast between the elements are tunable.  Also, if the coherent tender to soft X-ray beams are obtained chemical XPCS might be realized and reveal various element specific interaction of function

Round 2

Reviewer 1 Report

The authors have adequately addressed all issues.